# Spatiotemporally precise optogenetic activation of sensory neurons in freely walking *Drosophila*

Brian D DeAngelis[1†], Jacob A Zavatone-Veth[2†‡], Aneysis D Gonzalez-Suarez[1], Damon A Clark[1,2,3,4]*

[1]Interdepartmental Neuroscience Program, Yale University, New Haven, United States; [2]Department of Physics, Yale University, New Haven, United States; [3]Department of Molecular, Cellular and Developmental Biology, Yale University, New Haven, United States; [4]Department of Neuroscience, Yale University, New Haven, United States

*For correspondence:
damon.clark@yale.edu

[†]These authors contributed equally to this work

Present address: [‡]Department of Physics, Harvard University, Cambridge, United States

Competing interests: The authors declare that no competing interests exist.

**Abstract** Previous work has characterized how walking *Drosophila* coordinate the movements of individual limbs (DeAngelis et al., 2019). To understand the circuit basis of this coordination, one must characterize how sensory feedback from each limb affects walking behavior. However, it has remained difficult to manipulate neural activity in individual limbs of freely moving animals. Here, we demonstrate a simple method for optogenetic stimulation with body side-, body segment-, and limb-specificity that does not require real-time tracking. Instead, we activate at random, precise locations in time and space and use post hoc analysis to determine behavioral responses to specific activations. Using this method, we have characterized limb coordination and walking behavior in response to transient activation of mechanosensitive bristle neurons and sweet-sensing chemoreceptor neurons. Our findings reveal that activating these neurons has opposite effects on turning, and that activations in different limbs and body regions produce distinct behaviors.

## Introduction

Recent advances in computer vision have provided increasingly detailed descriptions of *Drosophila* behavior (*Bidaye et al., 2019*; *DeAngelis et al., 2019*; *Günel et al., 2019*; *Mathis et al., 2018*; *Pereira et al., 2019*). To connect these characterizations of motor output to neural circuitry, it is necessary to simultaneously manipulate neural activity and observe behavior (*Krakauer et al., 2017*). Various studies have manipulated neural activity and circuitry and measured locomotor phenotypes (*Berman et al., 2014*; *Bidaye et al., 2014*; *Bidaye et al., 2019*; *Cande et al., 2018*; *DeAngelis et al., 2019*; *Hampel et al., 2015*; *Hampel et al., 2017*; *Martin et al., 2015*; *Mathis et al., 2018*; *Mendes et al., 2013*; *Pereira et al., 2019*; *Seeds et al., 2014*; *Strauss and Heisenberg, 1990*; *Zhang et al., 2020*), but these studies have focused primarily on simultaneously manipulating the activity of all neurons in a class. However, behaviors are guided by patterns of neural activity that are precise in both time and space. In particular, many behaviors are asymmetric in nature, such as the anti-phasic coordination of limbs during walking (*Bidaye et al., 2014*; *Bidaye et al., 2019*; *DeAngelis et al., 2019*; *Mathis et al., 2018*; *Mendes et al., 2013*; *Pereira et al., 2019*; *Strauss and Heisenberg, 1990*), the asymmetrical postures associated with turning (*DeAngelis et al., 2019*; *Martin et al., 2015*; *Strauss and Heisenberg, 1990*), and grooming (*Berman et al., 2014*; *Hampel et al., 2015*; *Seeds et al., 2014*; *Zhang et al., 2020*). It has been difficult to generate neural manipulations that target specific neurons within a genetically identified class in freely moving insects. Here, we present a simple method for spatiotemporally precise activation of neurons in intact, freely moving animals. Using this method, we characterize the behavioral

responses of walking adult fruit flies to spatiotemporally restricted activation of peripheral mechano-sensory and chemosensory neurons. Our results reveal the asymmetries and region-specificity of behavioral programs recruited by these two types of sensory neurons.

Light sensitive channels allow neural activity to be manipulated with millisecond temporal resolution (*Boyden et al., 2005*). In *Drosophila*, large libraries of cell-specific drivers allow researchers to target specific, genetically-identifiable classes of neurons (*Dionne et al., 2018*; *Gohl et al., 2011*; *Jenett et al., 2012*). However, to manipulate a circuit in a spatiotemporally precise manner, it is necessary either to express a genetic tool in an anatomically precise way or to deliver a spatially precise stimulus to a broader expression pattern. Early studies in *Drosophila* used gynandromorphs to create bilaterally differentiated flies (*Hotta and Benzer, 1970*; *Hotta and Benzer, 1972*). Alternatively, sparse, stochastic expression patterns can be generated across a population, which then can be screened to determine the expression patterns of individual specimens (*Bidaye et al., 2019*; *Lee and Luo, 1999*; *Ribeiro et al., 2018*; *Wu et al., 2016*; *Xu and Rubin, 1993*). Both approaches rely on time-intensive techniques. Recently, sophisticated optical techniques have been developed for spatially precise optogenetic or thermogenetic manipulation in *Drosophila* and in *C. elegans*. These methods required real-time tracking and targeting of light paths (*Bath et al., 2014*; *Leifer et al., 2011*; *Wu et al., 2014*), which can be both computationally and mechanically challenging.

Here, we describe a simple method for spatiotemporally precise manipulation of neurons in freely walking adult *Drosophila*. Using a digital light projector, we project short pulses of random spatial patterns of light within an arena, while simultaneously recording the behavior of freely walking flies. Post hoc, we localize the projected stimuli to body and limb positions. This method can manipulate neural activity with body-side-, body-segment-, and limb-specificity without computationally intensive real-time tracking. Using this technique, we activated two distinct classes of superficial sensory neurons: bristle mechanoreceptor neurons (*Tuthill and Wilson, 2016a*) and sweet-sensing neurons expressing the Gr5a receptor (*Dahanukar et al., 2007*). We reasoned that, since both neuron classes are used as sensors during navigation and food-seeking, activating either of them would likely alter walking patterns. We measured the behavioral responses of walking *Drosophila* to spatially localized impulses of activity in these two cell types. These experiments revealed distinct, body region-specific and limb-specific maps of evoked behavior, which show how patterned sensory inputs influence complex coordination tasks. In combination with sophisticated tools for markerless tracking, this approach can be applied to show how sensory inputs are integrated in time and space to control walking coordination and navigation.

## Results

### Post hoc matching permits localized stimulation without real-time tracking and targeting

To determine how anatomically restricted neural activations influence locomotor behavior, we developed a novel method for spatiotemporally precise optogenetic stimulation of freely walking flies. We extended our assay for measuring *Drosophila* locomotor behavior (*DeAngelis et al., 2019*) by mounting a projector below the walking substrate (*Figure 1A*, see Materials and methods). Using this setup, short pulses of spatially patterned light were projected onto the arena from below. Importantly, the spatial pattern, duration, and wavelength of the stimulus can be adjusted in the software used to control the projector (*Brainard, 1997*; *Kleiner et al., 2007*; *Pelli, 1997*). Relying on the stereotyped size of the fly's body and previously-developed tracking algorithms (*DeAngelis et al., 2019*), we determined post hoc when and where optogenetic stimulation had occurred during an experiment (*Figure 1B–C*, see Materials and methods). In our analysis, we excluded all cases in which multiple limbs, or a limb and the body, were stimulated simultaneously. We grouped activations based on whether they hit the head, thorax, abdomen, forelimb, midlimb, or hindlimb of the fly (*Figure 1C*, see Materials and methods). Because baseline forward walking velocity varies over time and between flies, we analyzed fold-changes from the pre-stimulus forward speed (*Creamer et al., 2018*; *DeAngelis et al., 2019*). We validated this method by showing that, in

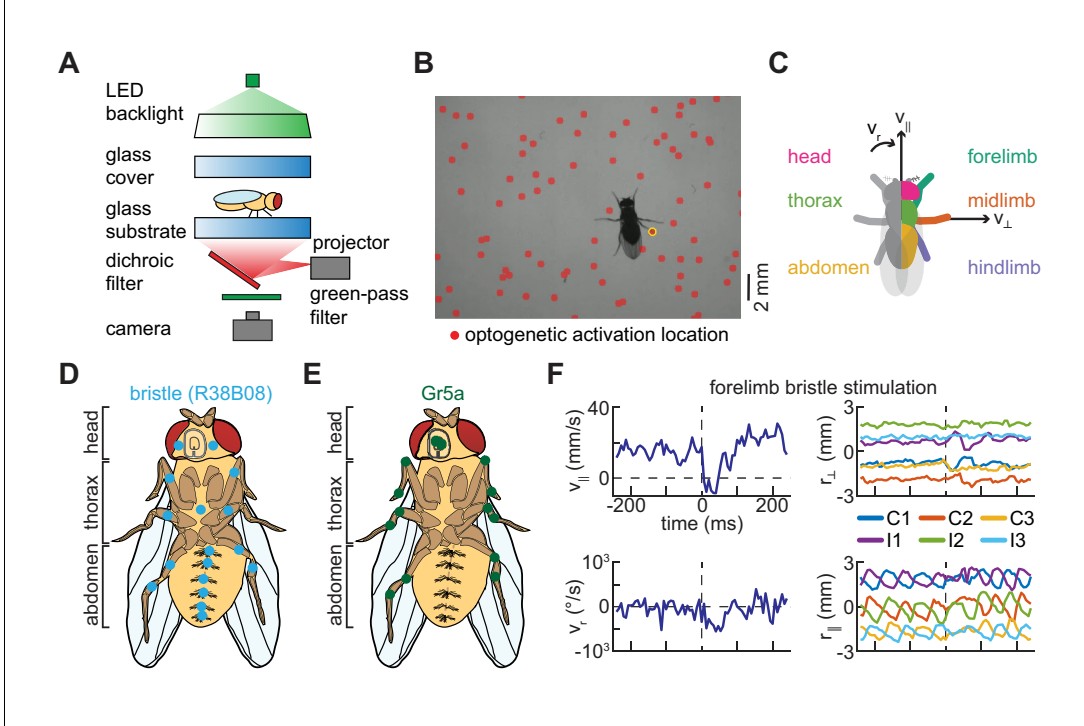

**Figure 1.** A simple method for spatiotemporally precise optogenetic manipulation in freely walking *Drosophila*. (A) Schematic of experimental setup for spatiotemporally precise optogenetic activation (see Materials and methods for details). Flies walk in a circular arena while illuminated from above and tracked from below using a high-speed camera. Localized optogenetic activations are generated using a digital light projector mounted below the arena. (B) Sample video frame showing optogenetic activation localized to a single limb. The spatially patterned stimulus is shown as a *red* overlay on the image, and the detected activation is circled in *yellow*. (C) Diagram of grouping of activations, showing the fly from above. Activations on the left and right sides of the body are symmetrized such that all appear on the right. Activations on the body are grouped by whether they hit the head (*magenta*), thorax (*green*), or abdomen (*yellow*) of the fly, while activations on the limbs are grouped by whether they hit the forelimb (*teal*), midlimb (*orange*), or hindlimb (*purple*). Black arrows indicate the three components of body movement: forward walking velocity ($v_{\parallel}$), lateral walking velocity ($v_{\perp}$), and heading angular velocity ($v_r$). (D) Schematic depiction of the distribution of bristle neurons across the limbs and ventral surface of the fly (see *Figure 1—figure supplement 1*; *Tuthill and Wilson, 2016a*). (E) Schematic depiction of the distribution of Gr5a-expressing (sweet-sensing) neurons across the limbs and ventral surface of the fly (*Dahanukar et al., 2007*; *Kwon et al., 2014*). (F) An example of turning evoked by optogenetic stimulation of forelimb bristles in freely walking flies. The *top left* panel shows the fly's forward walking speed as a function of time, the *bottom left* panel its yaw velocity, the *top right* panel the positions of its limbs in the direction perpendicular to its body axis, and the *bottom right* panel the positions of its limbs in the direction parallel to its body axis. In the right panels, C1, C2, and C3 indicate the fore-, mid-, and hind-limbs on the side of the body contralateral to the hit, while I1, I2, and I3 indicate the limbs on the ipsilateral side. See also *Figure 1—figure supplement 1*.

The online version of this article includes the following figure supplement(s) for figure 1:

**Figure supplement 1.** Expression patterns of the R38B08-Gal4 driver in bristle neurons.

flies that do not express optogenetic constructs, the stimulus alone does not significantly influence locomotor behavior. Overall, this approach is advantageous because it allows for non-invasive body side-, body segment-, or limb-specific perturbations without requiring computationally-intensive real-time tracking and targeting (*Bath et al., 2014*; *Leifer et al., 2011*; *Wu et al., 2014*).

## Mapping behavioral responses to impulse activations of peripheral sensory neurons

A fly's sense of touch is primarily mediated by bristles that cover the surface of its limbs and body (*Figure 1D*, see Materials and methods) (*Burrows, 1996*; *Tuthill and Wilson, 2016b*; *Tuthill and Wilson, 2016a*). These bristles are innervated by neurons that are genetically targetable across many parts of the fly's body (*Figure 1—figure supplement 1*). Mechanical perturbations of these bristles can generate region-specific behavioral responses (*Ramdya et al., 2015*). Whereas the activation of bristle neurons is thought to be an aversive stimulus (*Ramdya et al., 2015*; *Tuthill and Wilson, 2016a*), activation of sweet-sensing neurons that express the Gr5a chemoreceptor is thought to

**Table 1.** Experimental strains.

| Description | Genotype | Use | Figures |
|---|---|---|---|
| CsChrimson Control | +; +; UAS-CsChrimson | Control | *Figure 2—figure supplements 1, 2, Figure 3—figure supplements 1, 2, 3, Figure 4—figure supplement 2* |
| Bristle > CsChrimson | +; +; R38B08-Gal4/UAS-CsChrimson | Bristle activation | *Figures 2, 3, 4* |
| Gr5a > CsChrimson | +; Gr5a-Gal4/ +; UAS-GFP/UAS-CsChrimson | Gr5a activation | *Figures 2, 3, 4* |

be an appetitive stimulus (*Dahanukar et al., 2007*). Unlike bristle neurons, which are present across the entirety of the cuticle, neurons that express the Gr5a receptor are found primarily along the labellum and on each of the fly's six limbs (*Figure 1E*; *Kwon et al., 2014*). Given that activations of these two peripheral neuron types are thought to have opposite valances, we hypothesized that localized perturbations could evoke distinct, region-specific behavioral responses in freely-walking flies.

We applied our stimulation method to characterize the behavioral responses of freely walking flies to impulse activations of these two classes of peripheral sensory neurons. To do this, we drove the optogenetic activator CsChrimson (*Klapoetke et al., 2014*) in these neuron classes (see Materials and methods, *Table 1*) and monitored their behavior during stimulation. When these flies were briefly stimulated with random patterns of red light (see Materials and methods, *Table 2*), they displayed rich kinematic responses (*Figure 1F*), both slowing and turning. Since our method allows us to locally activate neurons expressing these drivers on any portion of the body, we first analyzed these responses by building somatic maps of responses to stimulation on the ventral surface of the fly. In control flies, stimulation did not evoke significant responses (*Figure 2—figure supplement 1*). Behavioral responses to bristle neuron activation partitioned the fly's body into four distinct quadrants (*Figure 2*). Flies slowed when bristles were activated on the anterior half of the fly's body, including the head and thorax (*Figure 2A,D*). In contrast, they sped up in response to activations of bristle neurons on the abdomen. When neurons on one side of the body were activated, flies turned away from the side of the hit and walked laterally away from the side of the hit (*Figure 2D–F,H*). These movements are consistent with escape behaviors to move away from an aversive stimulus. Unlike bristle activation, stimulation of flies expressing CsChrimson in Gr5a-neurons evoked slowing that was qualitatively similar for hits on all regions of the body (*Figure 2*). When flies were hit on one side of the body, they did not turn strongly or walk laterally (*Figure 2C,E,G,I*). They slowed in response to hits to the head, but slowed less in response to hits to the thorax and abdomen. These responses are consistent with the Gr5a expression pattern, which is strong in neurons in the head. The responses to hits in the thorax and abdomen may reflect activation of neurites in the ventral nerve cord or may reflect scattered light activating taste neurons elsewhere on the body.

Activating bristle neurons generally increased the variability of walking kinematics across the population of stimulated flies (*Figure 3—figure supplement 2*). Interestingly, even though stimulating sweet-sensing neurons in the head caused slowing, just like bristle activation, stimulating these neurons on the fly's head evoked slight decreases in variability, in contrast to the bristle neuron activations (*Figure 3—figure supplement 2*). This difference may well arise from the stronger slowing evoked by the sweet-sensitive neurons, which limits the possible degree of variability.

**Table 2.** Experimental groups.

| Description | Number of recording sessions | Total fly count | Stimulus duration | Interleave duration |
|---|---|---|---|---|
| CsChrimson Control | 3 | 45 | 30 ms | 0.5 s |
| Bristle > CsChrimson | 5 | 75 | 10 ms | 0.5 s |
| Gr5a > CsChrimson | 5 | 67 | 30 ms | 1 s |

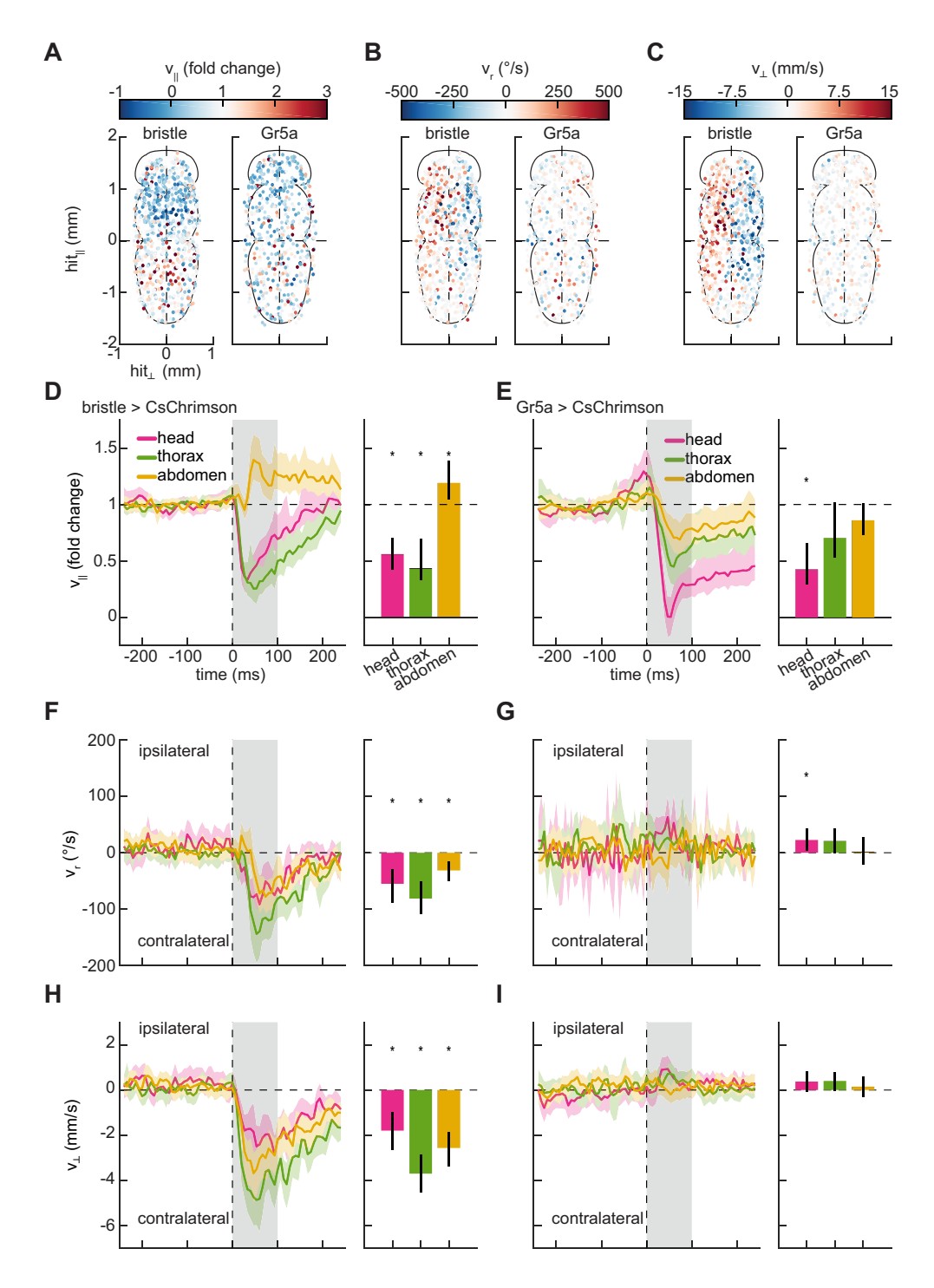

**Figure 2.** Mapping the responses of freely walking flies to optogenetic stimulation on the body. (**A**) Spatial maps of forward velocity responses to localized optogenetic stimulation of flies expressing bristle (*left*) and Gr5a-neuron (*right*) drivers at different locations on their bodies. Responses are averaged over the 100 ms following the onset of activation. Positive heading angular velocities and lateral velocities are rightward in this view. N = 785 bristle activations, and N = 568 sweet-sensor activations. (**B**) As in (**A**), but for heading angular velocity. (**C**) As in (**A**), but for lateral velocity. (**D**) Fold change in forward velocity as a function of time (*left*) and averaged over time (*right*) in response to optogenetic stimulation of bristle neurons on the head, thorax, or abdomen of the fly. At *left*, the *grey* patch indicates the window over which responses are averaged to generate the right panel, and error patches show 95% confidence intervals of the mean obtained from bootstrap distributions over activations. At *right*, black bars show 99% confidence intervals, estimated through bootstrapping, and significance stars show p<0.01 by a bootstrap permutation test against the null hypothesis
*Figure 2 continued on next page*

*Figure 2 continued*

that average forward velocity after activation is indistinguishable from that before activation. N = 158 head activations, 306 thorax activations, and 321 abdomen activations throughout. (E) As in (D), but for stimulation of flies expressing CsChrimson in Gr5a-neurons. N = 142 head activations, 179 thorax activations, and 247 abdomen activations throughout. (F) As in (D), but for the heading angular velocity of the fly. Positive heading angular velocities are directed towards the side of activation. (G) As in (F), but for stimulation of flies expressing CsChrimson in Gr5a-neurons. (H) As in (D), but for the lateral velocity of the fly. Positive lateral velocities are directed towards the side of activation. (I) As in (H), but for stimulation of flies expressing CsChrimson in Gr5a-neurons. See also *Figure 2—figure supplements 1–2*.

The online version of this article includes the following figure supplement(s) for figure 2:

**Figure supplement 1.** Control flies do not show a significant behavioral response to the optogenetic stimulus.

**Figure supplement 2.** Localized optogenetic stimulation on the body of the fly alters the variability in its walking kinematics.

## Activation of bristle and sweet-sensing neurons on the fly's limbs evoke opposite responses

Having shown that localized optogenetic stimulation evoked body-region-specific responses, we next asked whether responses to stimulation on the fly's different limbs were similarly differentiated. We therefore quantified slowing, turning, and lateral walking responses to impulse activations of each of the limbs. As observed for stimulation on the body, stimulation on the legs of control flies did not evoke significant responses (*Figure 3—figure supplement 1*). Stimulation of both bristle and sweet-sensing neurons on any of the limbs of the fly produced significant slowing (*Figure 3A–B*). Strikingly, while activation of bristle neurons on any of the fly's limbs evoked turning and side-stepping away from the activation site (*Figure 3C and E*, *Video 1*), activation of sweet-sensing neurons on the forelimbs or midlimbs evoked turning and sidestepping towards the activation site (*Figure 3D and F*). Activating bristle neurons consistently increased the response variability, as was observed for stimulation on the body (*Figure 3—figure supplement 2*). In contrast, depending on the stimulated limb, activating sweet-sensing neurons evoked either increases or decreases in kinematic variability (*Figure 3—figure supplement 2*), consistent with potential floor effects due to slowing that limit variability. The opposing responses to bristle and sweet-sensing neuron activations in the limbs are consistent with the hypothesized opposing valances of activity in these sensory neurons.

Because the limbs move periodically, one might expect that behavioral responses to activity in these neurons might depend on limb phase. To test this hypothesis, we separately analyzed hits that occurred during each limb's swing and stance phases (*Figure 3—figure supplement 3*). The activation construct CsChrimson remains active for ~20 ms in vitro (*Klapoetke et al., 2014*). Since steps take ~100 ms or longer, our methodology can in principle generate phase-specific activations, though the activation kinetics may vary across neuron types and expression levels in vivo. For both bristle and sweet-sensing neurons, average turning responses did not depend strongly on whether the limb was in stance or in swing during activation. Stimulating limbs during stance phase tended to evoke a greater degree of slowing than stimulating during swing phase, particularly for activation of sweet-sensing neurons on the midlimbs. This effect was also present to some degree in control flies (*Figure 3—figure supplement 3*), which likely reflects the fact that the fly's walking speed varies with phase (*DeAngelis et al., 2019*). Some of the observed discrepancy could therefore be attributable to sampling bias (*Figure 3—figure supplement 3*, see Materials and methods). Because bristle and sweet neuron activations both evoke substantial overall slowing, it is difficult to rigorously compare phase-dependent differences in responses to stimulation to the underlying phase dependence of the walking behavior and strong slowing. However, our data suggest that bristle neuron activation evokes similar behavioral responses irrespective of whether the limb is stimulated during swing or stance, while sweet neuron activation may evoke stronger slowing responses when stimulation occurs while the limb is in contact with the substrate.

## Localized activations evoke changes in walking geometry

Last, we sought to quantify how activating sensory neurons on the fly's limbs changed their limb coordination. Limb coordination can be complex (*DeAngelis et al., 2019*), so a simple metric is the mean distance between contralateral limbs. During smooth walking, the mean distances between contralateral limbs are relatively constant when averaged over phase (*DeAngelis et al., 2019*;

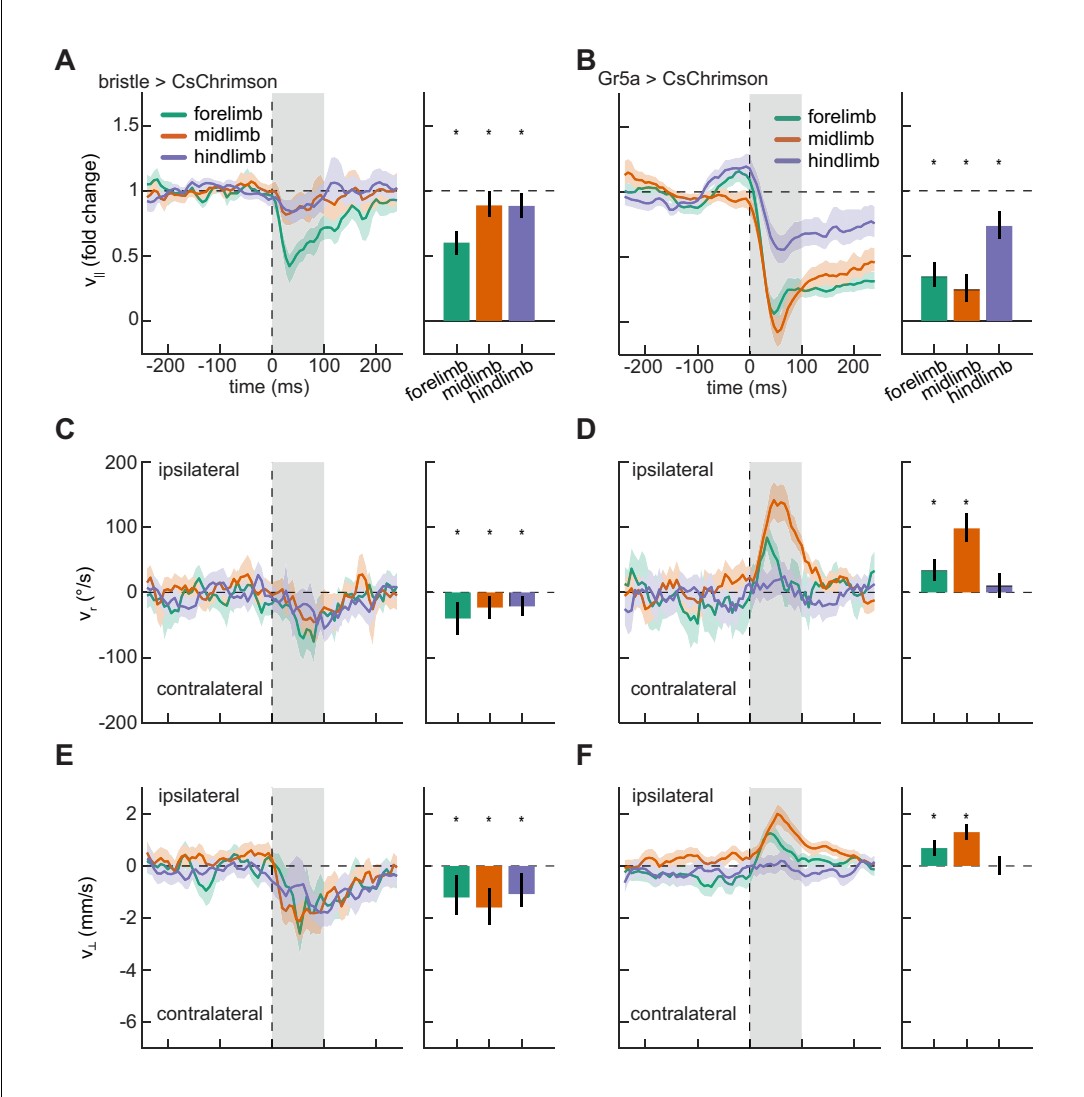

**Figure 3.** Localized activation of bristle and sweet-sensing neurons evokes opposing limb-specific responses. (A) Fold change in forward velocity as a function of time (*left*) and averaged over time (*right*) in response to optogenetic stimulation of bristle neurons on the forelimb, midlimb, or hindlimb of the fly. At *left*, the *grey* patch indicates the window over which responses are averaged to generate the right panel, and error patches show 95% confidence intervals of the mean obtained from bootstrap distributions over activations. At *right*, black bars show 99% confidence intervals, estimated through bootstrapping, and significance stars show p<0.01 by a bootstrap permutation test against the null hypothesis that average forward velocity after activation is indistinguishable from that before activation. N = 285 forelimb activations, 329 midlimb activations, and 324 hindlimb activations throughout. (B) As in (A), but for stimulation of sweet-sensing neurons. N = 307 forelimb activations, 508 midlimb activations, and 379 hindlimb activations throughout. (C) As in (A), but for the heading angular velocity of the fly. Positive heading angular velocities are directed towards the side of activation. (D) As in (C), but for stimulation of sweet-sensing neurons. (E) As in (A), but for the lateral velocity of the fly. Positive lateral velocities are directed towards the side of activation. (F) As in (E), but for stimulation of sweet-sensing neurons. See also *Figure 3—figure supplements 1–3* and *Video 1*.

The online version of this article includes the following figure supplement(s) for figure 3:

**Figure supplement 1.** Control flies do not show a significant behavioral response to optogenetic stimulation on their limbs.

**Figure supplement 2.** Localized optogenetic stimulation on the limbs of the fly alters the variability in its walking kinematics.

**Figure supplement 3.** Responses conditioned on whether the stimulated limb was in swing or stance phase.

---

*Mendes et al., 2013*; *Strauss and Heisenberg, 1990*). However, during spontaneous turning, the fly's forelimbs tend to move further apart, while the mid and hindlimb remain at roughly the same distance (*Figure 4—figure supplement 1*), consistent with forelimbs 'reaching' away from the body axis during turns (*DeAngelis et al., 2019*). The distances between limbs are therefore a proxy for

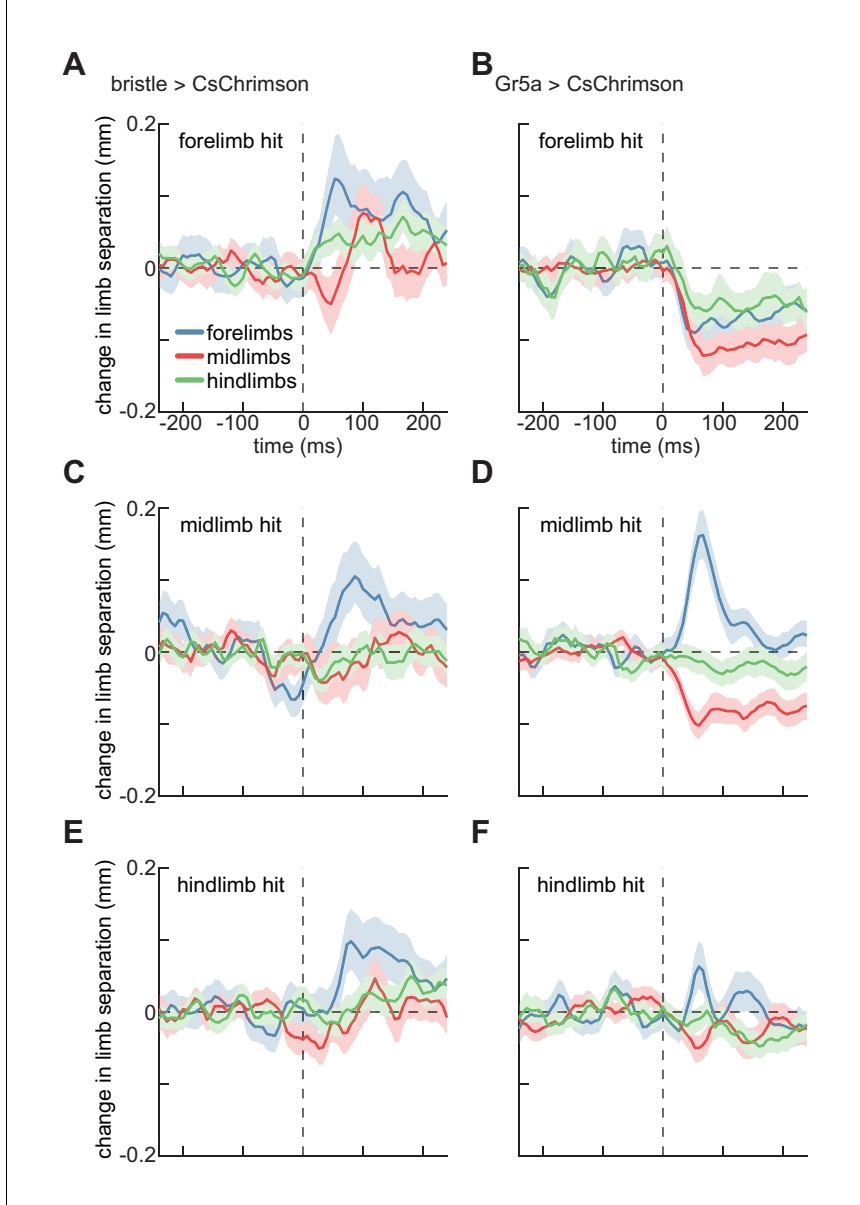

**Figure 4.** Localized optogenetic stimulation evokes changes in walking geometry. (**A**) Timeseries of change in Euclidean distance between contralateral limbs before and after activation of bristle neurons on the forelimb. Error patches show 95% confidence intervals of the mean, estimated through bootstrapping (N = 285 activations). (**B**) As in (**A**), but for activation of sweet-sensing neurons (N = 307 activations). (**C**) As in (**A**), but for activations on the fly's midlimb (N = 329 activations). (**D**) As in (**C**), but for activation of sweet-sensing neurons (N = 508 activations). (**E**) As in (**A**), but for activations on the fly's hindlimb (N = 324 activations). (**F**) As in (**E**), but for activation of sweet-sensing neurons (N = 379 activations). See also *Figure 4—figure supplements 1–2*.

The online version of this article includes the following figure supplement(s) for figure 4:

**Figure supplement 1.** Limb separation changes during spontaneous turning.

**Figure supplement 2.** Control flies do not show significant changes in the separation of contralateral limbs.

limb coordination, which can be complex during disrupted walking (*DeAngelis et al., 2019*). Moreover, limb-specific stance widening has been measured in stationary locusts when sensory bristles are mechanically stimulated (*Burrows, 1996*; *Laurent, 1986*; *Siegler and Burrows, 1986*). We therefore quantified changes in walking geometry by the separation between contralateral limbs after stimulation.

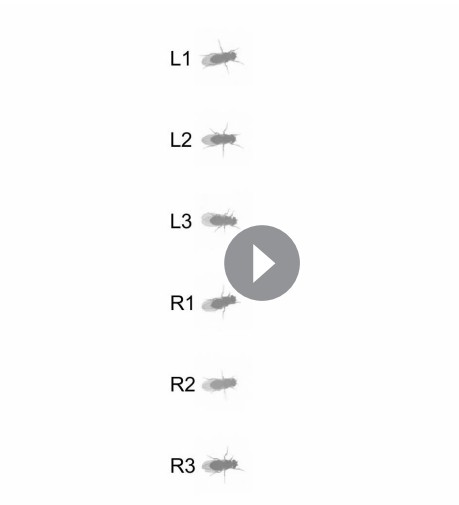

**Video 1.** Movie showing example responses to optogenetic stimulation of bristle neurons on each of the fly's limbs. Red circles indicate the location of the activation in space and time.
https://elifesciences.org/articles/54183#video1

We did not observe any effects on this coordination metric in control flies (*Figure 4—figure supplement 2*). Activating bristle neurons on the fly's forelimbs evoked a significant increase in the separation between its forelimbs, and a significant, though smaller, increase in the separation between the hindlimbs (*Figure 4A*). In contrast, stimulation of sweet-sensing neurons on the fly's forelimbs evoked decreases in the separation of all contralateral limb pairs (*Figure 4B*). Both bristle and sweet-sensor activation on the fly's midlimbs or hindlimbs evoked increases in the separation between contralateral forelimbs (*Figure 4C–F*). However, sweet-sensor stimulation evoked slight decreases in midlimb separation, which were not observed after bristle stimulation. (*Figure 4C–F*). In many of these cases, the changes in limb separation are not segment specific, suggesting more complex interactions among limbs than reported for stationary locusts (*Burrows, 1996*; *Laurent, 1986*; *Siegler and Burrows, 1986*). The fact that the separation between contralateral forelimbs also increases during spontaneous turning (*Figure 4—figure supplement 1*), suggests that some of these changes may be linked to the observed stimulus-evoked turning. Interestingly, only the forelimbs show strong modulation during spontaneous turning, while impulse activations of sensory neurons cause modulations in mid- and hindlimbs as well. Thus, these impulses likely evoked atypical coordination patterns.

## Discussion

In this study, we present a novel method for spatiotemporally localized optogenetic stimulation in freely moving animals (*Figure 1*). Using this method, we investigated the behavioral response of walking *Drosophila* to spatially restricted activation of bristle mechanoreceptor neurons and Gr5a-expressing taste neurons. Activating bristle neurons caused the fly to turn away from the site of activation, while activation of Gr5a neurons evoked turning towards the site of activation (*Figures 2* and *3*). Stimulation of both classes of neurons altered the walking geometry (*Figure 4*). The organization of the observed region- and limb-specific responses are consistent with the hypothesized navigational valences of these peripheral sensory neurons. These results show how flies use a suite of sensory inputs to make moment-to-moment decisions about heading and walking speed.

### Optogenetic manipulations disassociate neural and mechanical impulses

Using optogenetic methods for neural perturbations has several advantages over alternatives. First, though one can mechanically perturb a freely moving insect (*Jindrich and Full, 2002*; *Ristroph et al., 2010*), doing so necessarily imparts a mechanical impulse to the animal, confounding attempts to isolate neuronal effects. Similarly, sweet-sensing neurons may be activated by presenting a sugar droplet (*Shiraiwa and Carlson, 2007*), but doing so activates several classes of chemosensory and mechanosensory neurons. Second, previous studies have stimulated populations of neurons using implanted electrodes, and applied post hoc matching to identify the stimulated neurons (*Martin et al., 2015*). However, such preparations require invasive surgical procedures and may non-specifically stimulate unintended neuron classes. In contrast to these approaches, localized optogenetic stimulation is noninvasive, can be highly neuron-class-specific, and is easily applied to many locomotory modalities.

## Limitations of this method

This method has several limitations. First, because activations are identified post hoc, generating sequenced patterns of activity would require clever stimulus design. Second, though this method elicits strong behavioral responses with low-intensity light in peripheral structures (the limbs), scattering and absorption likely reduce the effectiveness of this technique for precisely activating deeper structures of the nervous system. We see evidence that such scattering may also reduce the specificity of this method, for instance in the mild responses to stimulation on the abdomen in flies expressing CsChrimson in Gr5a-neurons (*Figure 2E*). Third, though we activate using random spatial patterns, there are biases in the phase of activation because limbs spend more time in stance than in swing on average, and because limbs can intercept an activation patch as they swing forward. Despite these limitations, as the range of neuron classes accessible with genetic tools in *Drosophila* expands (*Cande et al., 2018*; *Luo et al., 2018*; *Mamiya et al., 2018*; *Namiki et al., 2018*), the ability to perform spatiotemporally precise optogenetic stimulation is particularly useful. Locomotor behaviors in *Drosophila* represent an opportunity to combine high-throughput, detailed behavioral characterization with genetic tools to dissect the circuit bases for distinct motor programs.

## Impulse responses in behavior

Beyond the experimental limitations enumerated above, it is important to highlight a fundamental limitation of optogenetic manipulations. While optogenetic perturbations allow us to manipulate neural activity with millisecond resolution, the method presented, like many optogenetic manipulations, focuses on impulse perturbations (*Cande et al., 2018*; *DeAngelis et al., 2019*; *Gepner et al., 2015*; *Hernandez-Nunez et al., 2015*). While understanding impulse responses is important, such manipulations do not generally recreate the spatiotemporal patterns of activity present during ethologically relevant behaviors. Generating activation patterns that mimic naturalistic activity in neural circuits remains difficult for several reasons. First, fully characterizing the patterned activity associated with naturalistic behaviors requires extensive neurophysiological experiments. Second, even with a complete description of relevant neural activity, generating naturalistic patterns of activity across a heterogenous population of neurons would require significantly more sophisticated techniques than currently possible in a non-invasive preparation. Thus, the transient activations generated with this and other optogenetic methods are unlikely to mimic the naturalistic activity of targeted neurons. Given this limitation, caution should be taken when relating observed phenotypes to naturalistic behaviors. Nonetheless, it is also clear that behavioral impulse responses to neural activation can shed significant light on the function of neural circuits (*Bidaye et al., 2014*; *Cande et al., 2018*; *DeAngelis et al., 2019*; *Gepner et al., 2015*; *Hernandez-Nunez et al., 2015*; *Martin et al., 2015*; *Robie et al., 2017*).

## Limb- and body-segment-specific mechanosensory modulation of behavior

In insects, mechanosensory inputs are involved in a wide range of behaviors including courtship, interactions with other insects, feeding, grooming, and locomotion (*Burg et al., 1993*; *Ejima and Griffith, 2008*; *Ramdya et al., 2015*; *Zhang et al., 2020*; *Zhou et al., 2019*). Previous work showed that tactile interactions facilitate group avoidance of aversive odor stimuli through stereotyped locomotor responses to distal limb touch (*Ramdya et al., 2015*). In this manuscript, we expanded on this understanding. We mapped behavioral responses to bristle neuron activation on the body of the fly (*Figure 2*). This map revealed both stereotypy in locomotor responses as well as sharp anatomical boundaries between accelerations and decelerations, as well as left and right turns. The spatial organization of these responses is consistent with bristle activation being an aversive stimulus, since in each of its four body quadrants, the fly directed its movement such that it increased the distance between the locus of activation and its center of mass. Similarly, when bristle neurons were activated in the limbs, flies slowed, turned away from the side of activation, and walked laterally away from the side of activation, all consistent with stimulus avoidance (*Figure 3*). Interestingly, the phase of the activation of the limb (stance vs. swing) had little apparent effect on the evoked behavior (*Figure 3—figure supplement 3*). Somatotopic mapping provides a powerful tool for investigating the neural implementation of sensory feedback during locomotion.

## Limb-specific chemosensory modulation of behavior

A striking feature of insect anatomy is the distribution of chemosensory receptors across the cuticle surface, including the limbs (*Kwon et al., 2014*). In *Drosophila*, 68 Gustatory Receptors (Grs) have been identified, with 28 Gr genes expressed in the limbs (*Kwon et al., 2014*). The projection patterns of all Gr-expressing neurons have been characterized in both the brain and VNC of the fly (*Kwon et al., 2014*). Among the Gr-expressing neurons in the legs, only those that express Gr5a both enervate all six leg neuropils in the VNC and do not send projections anteriorly towards the brain (*Dahanukar et al., 2007*; *Kwon et al., 2014*).

Our mapping of behavioral responses to activations of Gr5a neurons within the limbs revealed distinct, limb-specific phenotypes (*Figures 3* and *4*). The observed slowing in response to activation of Gr5a neurons is consistent with an appetitive stimulus, since slowing increases the time spent in contact with the location of activation and may be a preparatory behavior before tasting the surface. Slowing responses may also reflect antagonism between walking and feeding behaviors (*Mann et al., 2013*). It is interesting that activating Gr5a-expressing neurons in the head can elicit behavioral responses even when the fly has no evidence proboscis taste neurons are in contact with a substrate (*Figure 2*). This contrasts with the observation that flies appear to slow more when sweet-sensing neurons are activated when in contact with the substrate than while swinging (*Figure 3—figure supplement 3*). This result suggests the possibility of phase-specific gating of the influence of sensory stimuli, which seems ethologically sensible. Furthermore, the fact that activating these neurons evokes turning towards the stimulus indicates that the fly's food-seeking behavioral program depends on the limb activated, not merely the presence of a sweet taste. One intriguing possibility, given the lack of directly ascending projections from sweet-sensing neurons to the brain, is that these limb responses are mediated entirely by VNC circuitry. This would be consistent with the hypothesis that many complex insect behaviors are implemented in local sensorimotor loops (*Braitenberg, 1986*).

## A general method for spatiotemporally precise optogenetic stimulation in freely moving animals

The ability to activate neurons with high spatiotemporal precision has applications across neuroscience. Advances in automated behavioral tracking mean that this method could be used to compare responses between regions smaller than entire limbs (*Günel et al., 2019*; *Mathis et al., 2018*; *Pereira et al., 2019*). The simplicity of this method means that it could be applied broadly in small model organisms where optogenetic tools for non-invasive circuit manipulation are available. It is immediately applicable to *Drosophila* larvae, where non-localized stochastic optogenetic perturbations have been used to characterize navigational dynamics (*Gepner et al., 2015*; *Hernandez-Nunez et al., 2015*). It could also be used to study recovery from destabilizing neural perturbations in flying *Drosophila* (*Ristroph et al., 2010*), where real-time tracking can be difficult. In *C. elegans*, simultaneous optogenetic activation and measurement of whole-brain activity during behavior is an active area of research (*Leifer et al., 2011*; *Nguyen et al., 2015*). In larval zebrafish, asymmetric behaviors are studied primarily in tethered experimental preparations (*Naumann et al., 2016*; *Portugues et al., 2013*), but this method could provide a simple way to spatially target neural activation in freely swimming larvae. Simple methods for non-invasive, spatiotemporally precise circuit manipulation, like the method presented here, will aid in elucidating the neural bases of behavior.

## Materials and methods

### Fly strains and husbandry

As in prior work (*DeAngelis et al., 2019*), flies used in optogenetic experiments were grown at 25°C in a 12 hr/12 hr light/dark cycle and staged on $CO_2$ 0–6 days after eclosion. When staged, all flies used, including controls, were transferred to food supplemented with all-trans-retinal (ATR) following previous protocols (*De Vries and Clandinin, 2013*). Flies remained on ATR-supplemented food for four days prior to behavioral experiments. In all cases, flies were grown and experiments were performed at ~50% relative humidity.

## Experimental setup

Behavioral experiments were performed as in prior work (*DeAngelis et al., 2019*). Briefly, we used a 5 cm diameter circular planar arena consisting of two plates of glass separated by 2.5 mm (*Figure 1A*). The top glass plate was coated with Rain-X wax (Illinois Tool Works, Glenview, IL, USA) to prevent flies from walking on this surface during experiments. Above the arena, we mounted a diffusing screen and a 530 nm green Luxeon SP-01-G4 LED (Quadica Developments Inc, Lethbridge, AB, Canada) that provided background illumination (~1 µW/mm²). Flies were recorded from below at 150 fps using a Point Grey Flea3 FL3-U3-13Y3M-C camera (FLIR Systems, Wilsonville, OR, USA) fitted with a 25 mm C-mount lens (Tamron, Saitama City, Saitama Prefecture, Japan). The camera was positioned such that its field of view (2.2 × 2.75 cm) was approximately centered within the arena. The camera resolution was 0.043 mm/pixel. To increase the frequency of fly walking, all experiments were performed at 34° C (*Creamer et al., 2018*; *DeAngelis et al., 2019*; *Soto-Padilla et al., 2018*). During experiments, groups of 9–16 female flies were loaded into the arena and allowed to acclimate for 20 min prior to image acquisition. We then recorded the activity of flies in the arena for 1.1 hr, and tracked centroid and limb positions as in prior work (*DeAngelis et al., 2019*). Data from multiple recordings were merged to generate the aggregate datasets analyzed in this manuscript (see *Table 2*).

## Spatiotemporally precise optogenetic manipulations

A diagram of our physical setup for performing spatiotemporally localized optogenetic perturbations is shown in *Figure 1A*. As we were most interested in red-sensitive optogenetic constructs, we modified the behavioral experiment apparatus used in prior work (*DeAngelis et al., 2019*) by mounting a dichroic shortpass filter (Edmund Optics, Barrington, NJ, USA) at a 45° angle below the glass walking substrate. We added a green-pass filter (Semrock, Rochester, NY, USA) above the high-speed monochrome camera, so that we imaged only in green. Then, using a Lightcrafter 4500 digital light projector (Texas Instruments, Dallas, TX, USA), we projected patterned stimuli of red light onto the flies in the arena from below. With this geometry, all activations were on the ventral surface of the fly's body or limbs. We projected at 1440 Hz, and all activation durations were rounded to the nearest smaller integer number of frames. For the CsChrimson activation experiments presented in this work, we used red light with an intensity of ~0.05 mW/mm² and a central wavelength of 624 nm.

Stimuli were designed in Matlab using Psychtoolbox (*Brainard, 1997*; *Kleiner et al., 2007*; *Pelli, 1997*), using methods similar to those for visual or optogenetic stimulation of tethered flies (*Creamer et al., 2019*). We defined the stimulus in the frame of the camera and generate the appropriate bitmap for projection using a quadratic mapping between the coordinate system of the camera and that of the projector. Briefly, we let $\mathbf{x}_{cam}$ and $\mathbf{y}_{cam}$ be the vectors listing the coordinates of $N$ manually aligned reference pixels in the camera coordinate system, and $\mathbf{x}_{pro}$ and $\mathbf{y}_{pro}$ be the analogous vectors in the projector coordinate system, and defined the $N \times 6$ Vandermonde matrix of camera frame coordinates

$$\mathbf{X} = [1, \ \mathbf{x}_{cam}, \ \mathbf{y}_{cam}, \mathbf{x}_{cam} \circ \mathbf{x}_{cam}, \mathbf{x}_{cam} \circ \mathbf{y}_{cam}, \mathbf{y}_{cam} \circ \mathbf{y}_{cam}]$$

and the $N \times 3$ Vandermonde matrix of projector frame coordinates

$$\mathbf{Y} = [1, \ \mathbf{x}_{pro}, \ \mathbf{y}_{pro}]$$

where ° denotes the Hadamard (elementwise) product. Then, we fit the 6 × 3 transformation matrix $\mathbf{T}$ by solving the linear system

$$\mathbf{Y} = \mathbf{XT}$$

We can then use the transformation $\mathbf{T}$ to map each stimulus pixel from the camera coordinate frame to that of the projector. Note that because this transformation includes constant, linear, and quadratic terms, it can account for offsets, translations, rotations, stretching, and some degree of nonlinear warping between the two coordinate systems.

To perform spatiotemporally localized optogenetic stimulation, the general principle of our method is to present a random spatial pattern for a brief time, and then detect overlap between the activations and the insect post hoc after extracting centroid and limb positions using our tracking

algorithm. This approach is advantageous because it does not require computationally-intensive real-time tracking (*Bath et al., 2014*; *Leifer et al., 2011*; *Wu et al., 2014*). Here, we detected limb activations post hoc based on whether they intersect with a line segment between the center of mass of the fly and a given limb tip, outside an exclusion zone corresponding to the fly's body. For body activations, we relied on the stereotyped size of the fly to partition pixels into body regions. All instances in which more than one limb was stimulated were excluded from analysis, as were instances in which limbs were activated simultaneously with body hits. The percentage of all detected limb activations that were excluded based on simultaneous stimulation of more than one limb was 6%, 9%, and 7% for our bristle, Gr5a-neuron, and control datasets. In this study, we did not track wing location, so we did not exclude simultaneous hits of wings and limbs, or wings and body.

In the experiments described in this work, we used circular activation patches with a radius of 0.215 mm (five pixels in the coordinate system of the camera). The number of activation patches in each stimulus presentation was Poisson-distributed with parameter $\lambda = 35$ or 75 patches per frame (see *Table 2*). In each frame, activation patch locations were sampled as independent and identically distributed random variables with a uniform spatial distribution. (see *Figure 1C*). These stimuli were presented for 10 or 30 ms durations separated by 500 or 1,000 ms interleaves (see *Table 2*). We found that when the interleaves were shorter, or the density of activation spots was higher, flies tended to avoid the portion of the arena where we the activations took place. This was driver-dependent and occurred more for the activation of mechanosensory neurons. As our method did not track flies when they left the camera frame, which did not cover the entire arena, we did quantify potential behavioral adaptation to optogenetic stimulation. Thus, these stimulus parameters likely require tuning for specific drivers and experimental questions. These parameters also affect phase-dependent biases in hits, as the probability that a fly's limb will intercept an activation patch as it swings forward increases with the duration of each patch.

To analyze behavioral impulse responses, we aligned trajectory segments to detected activation onsets and averaged over many trials. Because baseline forward walking velocity varies over time and between flies, we analyzed fold-changes from the pre-stimulus forward speed (*Creamer et al., 2018*; *DeAngelis et al., 2019*). As we wished to focus our analysis on flies that were walking prior to activation onset, we excluded trajectories with average velocities slower than 3 mm/s in the 250 milliseconds preceding stimulation (*DeAngelis et al., 2019*). Data from multiple recording sessions were merged to generate the aggregate datasets analyzed in this manuscript (see *Table 2*).

## Confocal imaging of expression patterns

Confocal maximum projections of R38B08-Gal4 driving expression of green fluorescent protein (GFP) were obtained as follows. Female flies, aged between 7–10 days, were selected for GFP expression (fly genotype: w; UAS-MCD8-GFP/+; UAS-MCD8-GFP/R38B08-Gal4). Flies were sectioned; heads, pro-, meta-, and meso-thoracic legs, and abdomen/thorax were mounted in separate microscope slides. Decapitated heads were positioned ventral side up into wells on the microscope slides. The wells were then filled with paraffin oil and a coverslip was placed over the specimen. Similarly, severed abdomens and thoraces were mounted ventral-side up. All legs were isolated and mounted as described above. Body parts from four flies were imaged on different days. All flies were imaged using a Zeiss LSM 880 confocal microscope, using plan-apochromat 63x/1.40 NA and 40x/1.3 NA oil objectives. GFP was excited with a 488 nm laser and detected using a photomultiplier tube.

## Statistics

All statistical analysis was conducted using Matlab 9.6 and 9.7 (The MathWorks, Natick, MA, USA). Throughout this manuscript, presented error bars on time series are bootstrapped 95% confidence intervals obtained using the bias-corrected and accelerated percentile method (*Efron, 1987*). For integrated post-stimulus responses and corresponding significance testing, bootstrapped 99% confidence intervals are used to test for differences from baseline of mean centroid velocities after optogenetic activation.

## Data and code availability

The datasets of responses to optogenetic stimulation analyzed in this work are available from the Dryad Digital Repository: https://doi.org/10.5061/dryad.nzs7h44nk. The spontaneous turning data from *DeAngelis et al. (2019)* analyzed in *Figure 4—figure supplement 1* are also available from the Dryad Digital Repository: https://doi.org/10.5061/dryad.3p9h20r. Matlab code to generate projector stimuli, reproduce all statistical analyses, and generate all figure panels is available at https://github.com/ClarkLabCode/FlyLimbOptoCode (*Zavatone-Veth, 2020*; copy archived at https://github.com/elifesciences-publications/FlyLimbOptoCode).

## Acknowledgements

This work benefitted from many helpful conversations with M Venkadesan and O Mano. The Gr5a-Gal4 flies were a kind gift from J Carlson. We thank Y Luo and J Carlson for assistance with confocal microscopy. BDD was supported by an NSF GRF and a Gruber Science Fellowship. ADGS was supported by an NSF GRF and a Ford Foundation Predoctoral Fellowship. DAC and this project were supported by the Smith Family Foundation, NIH R01EY026555, NSF IOS1558103, a Searle Scholar Award, and a Sloan Fellowship in Neuroscience.

## Additional information

### Funding

| Funder | Grant reference number | Author |
| --- | --- | --- |
| National Institutes of Health | EY026555 | Brian D DeAngelis<br>Damon A Clark |
| National Institutes of Health | EY026878 | Brian D DeAngelis<br>Damon A Clark |
| Chicago Community Trust | Searle Scholar Award | Damon A Clark |
| Alfred P. Sloan Foundation | Fellowship | Damon A Clark |
| National Science Foundation | GRF | Brian D DeAngelis |
| Richard and Susan Smith Family Foundation | Scholar Award | Brian D DeAngelis<br>Damon A Clark |
| National Science Foundation | IOS 1558103 | Jacob A Zavatone-Veth<br>Damon A Clark |
| Gruber Foundation | Science Fellowship | Brian D DeAngelis |
| Ford Foundation | Predoctoral Fellowship | Aneysis D Gonzalez-Suarez |
| Searle Scholars Program | Scholar Award | Damon A Clark |

The funders had no role in study design, data collection and interpretation, or the decision to submit the work for publication.

### Author contributions

Brian D DeAngelis, Conceptualization, Data curation, Software, Investigation, Visualization, Methodology, Writing - original draft; Jacob A Zavatone-Veth, Conceptualization, Data curation, Software, Investigation, Visualization, Methodology, Writing - review and editing; Aneysis D Gonzalez-Suarez, Investigation, Visualization; Damon A Clark, Conceptualization, Supervision, Funding acquisition, Visualization, Methodology, Writing - review and editing

### Author ORCIDs

Brian D DeAngelis (iD) http://orcid.org/0000-0001-9418-7619
Jacob A Zavatone-Veth (iD) https://orcid.org/0000-0002-4060-1738
Damon A Clark (iD) https://orcid.org/0000-0001-8487-700X

Decision letter and Author response
Decision letter https://doi.org/10.7554/eLife.54183.sa1
Author response https://doi.org/10.7554/eLife.54183.sa2

## Additional files

### Supplementary files

• Transparent reporting form

### Data availability

Source data were deposited on Dryad: https://doi.org/10.5061/dryad.nzs7h44nk. Analysis code is available here: https://github.com/ClarkLabCode/FlyLimbOptoCode (copy archived at https://github.com/elifesciences-publications/FlyLimbOptoCode).

The following dataset was generated:

| Author(s) | Year | Dataset title | Dataset URL | Database and Identifier |
|-----------|------|---------------|-------------|-------------------------|
| DeAngelis BD, Za-vatone-Veth JA, Gonzalez-Suarez AD, Clark DA | 2020 | Data from: Spatiotemporally precise optogenetic activation of sensory neurons in freely walking *Drosophila* | https://doi.org/10.5061/dryad.nzs7h44nk | Dryad Digital Repository, 10.5061/dryad.nzs7h44nk |

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
