## [Decision Letter]

Thank you for submitting your article "Spatiotemporally precise optogenetic manipulation in freely moving animals" for consideration by *eLife*. Your article has been reviewed by three peer reviewers, and the evaluation has been overseen by Ronald Calabrese as the Senior Editor and Reviewing Editor. The reviewers have opted to remain anonymous.

The reviewers have discussed the reviews with one another and the Reviewing Editor has drafted this decision to help you prepare a revised submission.

Summary:

In this Advance to their recent *eLife* publication, DeAngelis and colleagues describe a new method for sparse optogenetic stimulation using a DLP projector. Specifically, they use the DLP system to randomly-scatter light at the wavelength necessary for optogenetic activation, and then, after the experiment, isolate the behavioral effects of stimulating as a function of where on the body the light hit. This method is cheaper and easier than more complicated methods that require real-time tracking and targeting for specific body parts. They first activate mechanosensory bristles in different parts of fly's body: stimulating the abdomen makes the fly speed up; stimulating the head, forelimbs, and thorax makes the fly slow down; activation on one side causes contralateral turning. They then activate sweet taste receptors (gr5a); here, flies tended to slow down when the head and body was stimulated, and slow down and turn toward the side of stimulation when legs were targeted. The paper qualifies nicely as an Advances article, as it produces novel results and methods important in directing future research.

Essential revisions:

While the reviewers were generally supportive there were concerns. First, the work appears presented more as a methods development than a scientific advance, leading to some confusion by the reviewers. Reviewer #1's confusion regarding the manuscript type would be best addressed by making clear that this is not a Tools and Resources paper but a scientific Advance. Second, the authors should more explicitly integrate this work with the previous publication – the continuous attractor dynamics play almost no role here. Some additional measurements (e.g., phase response curves, adaptation estimates, display of full variation rather than 95% c.i. of mean, etc.) would go a long way towards increasing the impact of the article. Reviewer #2's concerns should be fully addressed.

While reporting rigorous 95% CI's of the mean is certainly appropriate for many of the comparisons made some graphical presentation and quantification of the actual variation in the data would be useful for assessment of method applicability. We like the raw data plots of the fly body at the top of each figure, but it is hard to glean a number or distribution from this.

Reviewer #1:

In this Tools and Resources manuscript the authors build on their recent *ELife* publication to report a method for obtaining spatially and temporally selective optogenetic stimulation without real-time tracking or implanted devices. The key insight is to use patterned light and post-hoc analysis of high-speed video combined with extensively stochastic sampling. My major concerns are the lack of benchmarking and quantitative comparison to other methods and whether the methodology goes significantly beyond what might be appropriate for a Materials and methods section in a standard research article. The limitations of the proposed methods, common to many optogenetic approaches, are clearly articulated. The test cases are themselves a nice addition to the literature, enough so that this paper might be more appropriate as a research study in a more focused journal.

It is certainly a clever insight to use random sampling and post-hoc analysis to determine what category the stimulation fell under. However, this type of approach is certainly not unprecedented in other conditions such as multi-unit electrophysiology (e.g. Martin et al., 2015.). I have not seen it applied before in an optogenetics context, although I have not systematically looked, and it clearly does help in this situation. I think it quite evident that such method was useful in this case. However tools and techniques papers typical pose approaches that significantly improve upon existing data in a way that might resolve many open questions in the field. Combined with the lack of precise patterning of spikes, I am worried that the use cases here might be somewhat narrow.

*eLife*'s guidelines for a new method in a Tools and Resources papers says "…the new method should be properly compared and benchmarked against existing methods used in the field."

This seems like a reasonable request, especially for this paper. Specifically, I would very much like to see a quantification of precision and the degree of selectivity that this technique can provide. It clearly can distinguish between legs and body, but is that the limit? The figures suggest it may be more precise given the localization of the target, but how far outside of this region does the physiological activation extend? What are the typical levels of activation that are achieved when light passes through the exoskeleton? For the specific application here, these questions did not require answers because the authors examined the differential response between body and leg stimulation. However, for general methodology these characterizations seem necessary.

Reviewer #2:

In this update to their recent *eLife* publication, DeAngelis and colleagues describe a new method for sparse optogenetic stimulation using a DLP projector. They first activate mechanosensory bristles in different parts of fly's body: stimulating the abdomen makes the fly speed up; stimulating the head, forelimbs, and thorax makes the fly slow down; activation on one side causes contralateral turning. They then activate sweet taste receptors (gr5a); here, flies tended to slow down when the head and body was stimulated, and slow down and turn toward the side of stimulation when legs were targeted.

Overall, I think this short paper provides a reasonable update to the previous publication.

This update could be better integrated with the original publication. For example, the authors' previous submission contained a figure showing that activation of forelimb bristle neurons resulted in turns that appear distinct from spontaneous turning behavior. Including that data here would help to link these new results to those in the original paper.

It would be useful to construct a phase response curve of optogenetic stimulation to understand whether and how flies respond to perturbations at different phases of the step cycle. It would also be helpful to know when the leg is stimulated within the step cycle. It is likely that the random stimulation induces a bias in the phase that the leg gets stimulated at. In particular, I would expect that the legs get stimulated more when they are extended than when contracted.

For clarity, the authors need to provide more information about the distribution of bristle and gr5a neurons across the flies body. Without this information, it is difficult to interpret the behavioral results. Given that this is already described in the literature, it could be simply included on each figure in schematic form. For example, are there a similar number of Gr5a expressing neurons in each leg? What about on the abdomen? If not, why does the fly slow down when the abdomen is stimulated? Does this reveal something about the spatial resolution of the method?

Reviewer #3:

In this Advances article, the authors build upon their previous publication from last year, describing a method for analyzing the effects of location-specific optogenetic manipulations in a post hoc manner. Specifically, they use a DLP system to randomly-scatter light at the wavelength necessary for optogenetic activation, and then, after the experiment, isolate the behavioral effects of stimulating as a function of where on the body the light hit. This method is cheaper and easier than more complicated methods that require real-time tracking and targeting for specific body parts. As a proof of concept, the authors show sensible effects for stimulating two different types of neurons: bristle sensory neurons and Gr5a taste receptors.

On the whole, I think that this method, while having its limitations (all clearly stated in the Discussion), would be an important tool in the fly behavioral neuroscience toolbox, and I encourage acceptance of the article (especially as more of a methods paper). I am somewhat skeptical of the bristle mechanoreceptor results, as there appear to be non-specific targeting – looking back at the original paper – but I think that the results are compelling enough to serve as a proof-of-concept here. This is a caveat that I think needs to be made explicit in the final version, but, again, I think that this is a solid advance, and I encourage publication.

---

## [Author Response]

Essential revisions:While the reviewers were generally supportive there were concerns. First, the work appears presented more as a methods development than a scientific advance, leading to some confusion by the reviewers. Reviewer #1's comments are best addressed by making clear that this is not a Tools and Resources paper but a scientific Advance. Second, the authors should more explicitly integrate this work with the previous publication – the continuous attractor dynamics play almost no role here. Some additional measurements (e.g., phase response curves, adaptation estimates, display of full variation rather than 95% c.i. of mean, etc.) would go a long way towards increasing the impact of the article. Reviewer #2's concerns should be fully addressed.

The two essential revisions were (1) to make clear this is a scientific result, not just a tools paper and (2) to integrate this work better with our previous, related paper. To clarify the scientific contribution of this work and better connect it to our previous publication, we have extensively restructured our manuscript. Rather than presenting responses to bristle neuron and Gr5a-neuron activations sequentially, we now present these data in tandem, grouped by the location of stimulation. This organization better emphasizes the scientific insights generated by the activation method, and improves the overall clarity of the manuscript. Our title has also changed to emphasize the scientific content, rather than just the methodology.

Within this new framework, we have added several new figures to address questions raised by the reviewers, and which also more closely link the present results to our prior paper. First, we characterized the variability in evoked responses, showing that stimulation is followed by limb- and body-region-specific changes in kinematic variability across the population of stimulated flies (Figure 2—figure supplement 2 and Figure 3—figure supplement 2). Second, we now show how localized perturbations evoke changes in the geometry of limb coordination (Figure 4) and address the question of phase dependence in the responses (Figure 3—figure supplement 3). These two figures relate closely to the coordination patterns we studied in DeAngelis et al., 2019. Third, to address Reviewer 2’s question about the expression patterns of the R38B08 driver used to activate bristle neurons, we have acquired expression data, which show that this driver is expressed in neurons that appear to be associated with the sensory bristles on the ventral surface of the fly’s body (Figure 1—figure supplement 1).

While reporting rigorous 95% CI's of the mean is certainly appropriate for many of the comparisons made some graphical presentation and quantification of the actual variation in the data would be useful for assessment of method applicability. We like the raw data plots of the fly body at the top of each figure, but it is hard to glean a number or distribution from this.

To characterize the variability in responses, we have added two figures (Figure 2—figure supplement 2 and Figure 3—figure supplement 2), which show the mean absolute deviation of each of the components of the kinematics of the fly’s centroid as a function of time relative to stimulation onset. These plots quantify the variability in behavior and show that localized optogenetic stimulation evokes transient changes in the variability and stereotypy of kinematic behavior.

Reviewer #1:[…] It is certainly a clever insight to use random sampling and post-hoc analysis to determine what category the stimulation fell under. However, this type of approach is certainly not unprecedented in other conditions such as multi-unit electrophysiology (e.g. Martin et al., 2015.). I have not seen it applied before in an optogenetics context, although I have not systematically looked, and it clearly does help in this situation. I think it quite evident that such method was useful in this case. However tools and techniques papers typical pose approaches that significantly improve upon existing data in a way that might resolve many open questions in the field. Combined with the lack of precise patterning of spikes, I am worried that the use cases here might be somewhat narrow.eLife's guidelines for a new method in a Tools and Techniques papers says "…the new method should be properly compared and benchmarked against existing methods used in the field."This seems like a reasonable request, especially for this paper. Specifically, I would very much like to see a quantification of precision and the degree of selectivity that this technique can provide. It clearly can distinguish between legs and body, but is that the limit? The figures suggest it may be more precise given the localization of the target, but how far outside of this region does the physiological activation extend? What are the typical levels of activation that are achieved when light passes through the exoskeleton? For the specific application here, these questions did not require answers because the authors examined the differential response between body and leg stimulation. However, for general methodology these characterizations seem necessary.

Our primary response to this comment is included with our response to the Essential Revision. Determining the precise extent of physiological activation produced by optogenetic stimulation methods can be challenging, even when electrophysiological access is available (see, for instance, Li et al., 2019). The resolution with which one can target the optogenetic activation is limited by both the camera and projector resolutions. For the experiments presented here, the size of individual activation patches was 0.215 mm. The temporal precision of stimulation is limited both by the projector used (in this case, giving a maximum frame rate of 1440 Hz) and by the timescale over which the channels of the optogenetic activator remain open (~20 ms current decay time constant as measured in vitro for CSChrimson as used in this work, see Klapoetke et al., 2014). We would like to emphasize that many of these limitations are not inherent to the method; rather, they result from our implementation. In particular, the spatiotemporal precision of activation is constrained by the projector, camera, and optical configuration used. Given different equipment and a different stimulus generation protocol, one could achieve higher precision. There are, of course, limitations imposed by the scattering of the cuticle, which decreases resolution for non-superficial neurons, and could also generate non-specific activation, if scattered light can activate neurons that are quite distant to the spot of light. We now mention these limitations explicitly in the manuscript.

Reviewer #2:[…] This update could be better integrated with the original publication. For example, the authors' previous submission contained a figure showing that activation of forelimb bristle neurons resulted in turns that appear distinct from spontaneous turning behavior. Including that data here would help to link these new results to those in the original paper.

Our primary response to this comment is included with our response to the Essential Revision. Briefly, we have added visualizations to show how activation evokes changes in limb coordination.

It would be useful to construct a phase response curve of optogenetic stimulation to understand whether and how flies respond to perturbations at different phases of the step cycle. It would also be helpful to know when the leg is stimulated within the step cycle. It is likely that the random stimulation induces a bias in the phase that the leg gets stimulated at. In particular, I would expect that the legs get stimulated more when they are extended than when contracted.

Thank you for these suggestions. The reviewer’s expectation that random stimulation induces biases in the phase at which limbs are stimulated is well-founded, and we have revised the text to more clearly state this point. We have also added figures showing the distributions of limb phase at activation onset (Figure 3—figure supplement 3), which support these intuitions.

We agree that constructing phase response curves of optogenetic stimulation would be a useful and interesting analysis. However, to construct full phase response curves, one would ideally want a quantity of data sufficient to divide each cycle into at least four bins, ideally more. As we did not design our experiments with specific sample sizes in mind and did not perform any analysis until the full datasets were collected, we do not have enough data at our disposal to estimate phase response curves with high resolution.

To obtain a coarse-grained picture, we instead considered the mean responses to stimulation conditioned on whether the hit limb was in swing or stance phase at the time of stimulation (Figure 3—figure supplement 3). Under this analysis, average turning responses did not appear substantially phase-dependent. Stimulation of limbs during stance phase tended to evoke a greater degree of slowing than stimulation during swing phase, particularly for activation of Gr5a neurons on the flies’ midlimbs. However, control flies also displayed (qualitatively smaller) phase-dependent biases in walking speed, suggesting that this discrepancy could be at least partially accounted for by the sampling biases described above.

Yet, in this case, it remains unclear how to rigorously isolate possible phase-dependence in responses to stimulation from the underlying phase-dependence of slowing during walking behavior. One might seek to construct a shuffled internal control by resampling from the full dataset of stimulated flies. However, this is potentially problematic, due to the long timescale correlations present even in spontaneous fly walking behavior (see, for instance, Berman et al., 2016, Katsov et al., 2017, or Figure 1—figure supplement 2 of our previous *eLife* publication) and the possibility of longer-timescale behavioral effects of stimulation (such as the question of adaptation raised elsewhere). Beyond the fact that these effects make naïve resampling a poor control from a biological standpoint, they also can yield increased error rates in statistical tests (Politis et al., 1997, Canty et al., 2006). One could instead consider comparing against an external control, but one would then still have to contend with the issue of how closely pre-stimulation behavior should be matched. Furthermore, since these perturbations evoke substantial slowing, one would ideally want a control that accounts for potential phase-dependence in spontaneous slowing behavior. For these reasons, extracting a clear statistical picture of possible phase-dependence in responses is nontrivial, but an interesting avenue for future inquiry. We have revised the text to include these points, and we note the contrast in slowing for phase/swing hits in Gr5a midlimbs, but do not include statistical claims.

For clarity, the authors need to provide more information about the distribution of bristle and gr5a neurons across the flies body. Without this information, it is difficult to interpret the behavioral results. Given that this is already described in the literature, it could be simply included on each figure in schematic form. For example, are there a similar number of Gr5a expressing neurons in each leg? What about on the abdomen? If not, why does the fly slow down when the abdomen is stimulated? Does this reveal something about the spatial resolution of the method?

We have added a figure showing confocal imaging of superficial GFP-expressing cell bodies under the ventral cuticle of the fly (Figure 1—figure supplement 1), as well as diagrams showing the distribution of these neuron classes (Figure 1).

Reviewer #3:[…] On the whole, I think that this method, while having its limitations (all clearly stated in the Discussion), would be an important tool in the fly behavioral neuroscience toolbox, and I encourage acceptance of the article (especially as more of a methods paper). I am somewhat skeptical of the bristle mechanoreceptor results, as there appear to be non-specific targeting – looking back at the original paper – but I think that the results are compelling enough to serve as a proof-of-concept here. This is a caveat that I think needs to be made explicit in the final version, but, again, I think that this is a solid advance, and I encourage publication.

We thank the reviewer for their comments and hope that our revisions to the manuscript have further improved its clarity. In particular, as we note in our response to the Essential Revision, we have added imaging data to characterize the expression of the Gal4 driver lines used and have further expanded our discussion of the limitations of this method and of optogenetic impulse perturbations more generally.